# Contemporary Distribution, Estimated Age, and Prehistoric Migrations of Old World Monkey Retroviruses

Antoinette C. van der Kuyl 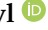

Laboratory of Experimental Virology, Department of Medical Microbiology, Amsterdam UMC,
University of Amsterdam, Meibergdreef 9, 1105 AZ Amsterdam, The Netherlands;
a.c.vanderkuyl@amsterdamumc.nl; Tel.: +31-205-666-778

**Abstract:** Old World monkeys (OWM), simians inhabiting Africa and Asia, are currently affected by at least four infectious retroviruses, namely, simian foamy virus (SFV), simian immunodeficiency virus (SIV), simian T-lymphotropic virus (STLV), and simian type D retrovirus (SRV). OWM also show chromosomal evidence of having been infected in the past with four more retroviral species, baboon endogenous virus (BaEV), Papio cynocephalus endogenous virus (PcEV), simian endogenous retrovirus (SERV), and Rhesus endogenous retrovirus-K (RhERV-K/SERV-K1). For some of the viruses, transmission to other primates still occurs, resulting, for instance, in the HIV pandemic. Retroviruses are intimately connected with their host as they are normally spread by close contact. In this review, an attempt to reconstruct the distribution and history of OWM retroviruses will be made. A literature overview of the species infected by any of the eight retroviruses as well as an age estimation of the pathogens will be given. In addition, primate genomes from databases have been re-analyzed for the presence of endogenous retrovirus integrations. Results suggest that some of the oldest retroviruses, SERV and PcEV, have travelled with their hosts to Asia during the Miocene, when a higher global temperature allowed simian expansions. In contrast, younger viruses, such as SIV and SRV, probably due to the lack of a primate continuum between the continents in later times, have been restricted to Africa and Asia, respectively.

**Keywords:** retrovirus; endogenous; exogenous; Old World monkey; BaEV; PcEV; SERV; SERV-K1; RhERV-K; SFV; SIV; SRV; STLV

## 1. Introduction

The primate fossil record suggests that the radiation of Old World monkeys (OWM) began in the Oligocene or early Miocene (around 25–30 million years ago (mya)), with the divergence of apes, including the ancestors of humans, and OWM (reviewed in [1], see also [2–4]). Subsequently, Colobinae and Cercopithecinae diverged around 16 mya, Papionini and Cercopithecini then split around 10–12 mya, and macaques and papionins separated approximately 7.6 ± 1.3 mya [1]. Over the course of evolution, retroviruses have been infecting primate species. Evidence for this can be found in primate genomes, where endogenous (i.e., germ line propagated) proviruses bear witness of past infections. The origin of the retrovirus family itself dates back to the early Paleozoic, approximately 460–550 mya [5]. Most likely, retroviruses originate from the long terminal repeat (LTR)-containing retrotransposons found in eukaryotic genomes, by the addition of an *envelope* (*env*) gene [6,7]. Indeed, one such LTR-retroelement, HERV-L, has similarity to foamy viruses in its *polymerase* (*pol*) gene [8].

At present, two retrovirus species infect humans, namely the lentivirus human immunodeficiency virus (HIV) and the deltaretrovirus human T-lymphotropic virus (HTLV). HIV has been transmitted from African non-human primates quite recently, probably around the beginning of the 20th century [9–11]. So far, five independent transmissions have been recorded, from chimpanzee and gorilla (HIV-1 group M, N, O, and P), and from sooty

mangabey (HIV-2), respectively [12–17]. There is, however, evidence that all eight HIV-2 subtypes in fact each represent independent transmissions of simian immunodeficiency virus from sooty mangabeys (SIVsm) [18–20], increasing the number of SIV transmission events considerably. Strains of HTLV have also been spreading from monkeys and apes to humans in the recent past, likely between 3000 and 378,000 years ago, depending on the particular virus type [21–28]. Moreover, transmissions of the virus are ongoing [29,30]. A third retrovirus, simian foamy virus (SFV), is likewise known for its ongoing, zoonotic transmissions from non-human primates to humans (reviewed in [31–33]), though a true human variant has not emerged yet. A further retrovirus, simian retrovirus (SRV), initially seen as the cause of simian AIDS in primate facilities, is largely an endemic pathogen of Asian macaques [34,35]. Serological evidence for incidental SRV infection in persons occupationally exposed to non-human primates has been reported [36].

For the existence of other simian retroviruses, namely, baboon endogenous virus (BaEV), Papio cynocephalus endogenous virus (PcEV), simian endogenous retrovirus (SERV) and rhesus endogenous retrovirus (RhERV-K/SERV-K1), all evidence we have comes from the "fossil" record. These viruses have disappeared from circulation but have left their proviral genomes in OWM chromosomes. All four appear to be OWM specific, as their sequences have not been found in hominoid genomes, and neither have novel infections in humans or apes been documented. Summarizing, the eight retroviral species listed here spread among African primates, belong to different retroviral families, and have various tendencies to target the germ-cell line. The exogenous primate retroviruses can all be transmitted to humans. This review and hypothesis paper will discuss, partly on the basis of new research, the epidemiology (who, where, and when?) and putative age of primate retroviruses from the Oligocene till present, including both the extant and the extinct members of the family, to deduce probable OWM retrovirus dispersal routes.

## 2. Materials and Methods

### 2.1. Survey Methodology

The PubMed database (www.ncbi.nlm.nih.gov/pubmed) and Google Scholar (scholar.google.nl) were manually searched using the name of the virus "simian immunodeficiency virus", "simian foamy virus", "simian T-lymphotropic virus", "Papio cynocephalus endogenous virus", "simian endogenous retrovirus", "baboon endogenous virus", or their abbreviation ("SIV", "SFV", "STLV", "BaEV", "PcEV", "RhERV", "SERV", respectively) and a second term appropriate to the topic, such as "age", "prevalence", "phylogeny", "evolution", "dating", "presence", "detection", "monkey", etc. Abstracts were inspected for relevance; from the selected abstracts, only those of which a full-length publication could be retrieved were read and included. In addition, references in the retrieved papers were inspected and used when appropriate.

### 2.2. Identifying Endogenous Retrovirus Sequences in OWM Genomes

PcEV (GenBank acc. no. AF142988 [37]) and SERV 23.1 (GenBank acc. no. U85505; [38]), both full-length proviruses obtained from a *Papio cynocephalus* (yellow baboon) chromosomal DNA library, were used to identify endogenous PcEV and SERV nucleotide sequences in OWM genomes from GenBank (www.ncbi.nlm.nih.gov/genome/) and Ensembl (www.ensembl.org/index.html) databases by similarity search using the BLASTn/BLAT algorithm provided. In addition, BaEV, another full-length provirus (GenBank acc. no. D10032; [39]), was used to search OWM genomes for BaEV integrations. CERV1 (GenBank acc. no. AY692036) and CERV2 (GenBank acc. no. AY692037) to search for CERV-like gammaretroviral sequences; SERV-K1 (GenBank acc. no. BK009405) and HERV-K5 (GenBank acc. no. DQ112093) were used to query OWM genomes for ERV-K family members.

Default settings were used in the NCBI BLASTn search, as the aim of the research was to retrieve and describe specifically PcEV, SERV and BaEV integrations, and not more distantly related endogenous viruses, searches were optimized for "highly similar sequences" (megablast). Only when no results were retrieved with megablast, the discontiguous

megablast option ("more dissimilar sequences") was used. In Ensembl BLAST searches, search sensitivity was the default ("normal") and filtering low complexity regions or filtering query sequences with RepeatMasker was disabled. When no full-length proviruses were retrieved, virus presence was defined by detecting fragments with >90% homology to both coding and non-coding (long terminal repeat, LTR) fragments of the specific virus. Primate genomes queried were: *Cercocebus atys, Cercopithecus mona, Cercopithecus neglectus, Chlorocebus sabaeus* (formerly known as *Cercopithecus aethiops sabaeus*), *Colobus angolensis, Erythrocebus patas, Macaca fascicularis, Macaca fuscata, Macaca mulatta, Macaca nemestrina, Mandrillus leucophaeus, Mandrillus sphinx, Papio anubis, Piliocolobus tephrosceles, Rhinopithecus bieti, Rhinopithecus roxellana, Theropithecus gelada,* and *Trachypithecus francoisi.*

### 2.3. Analysis of Proviral Sequences

BLAST results were downloaded from the databases and aligned using ClustalW as implemented in BioEdit (bioedit.software.informer.com). Alignments were optimized through visual inspection. Sequence distances were calculated with the Kimura-2-parameter method and evolutionary relationships were inferred using the Neighbor-joining method with bootstrapping as implemented in MEGA6 [40,41]. Gaps/missing data treatment was set to "partial deletion" with a cut-off value of 80%. No outgroup was defined. Alignments are available as Supplementary Files 1 and 2.

## 3. Results

### 3.1. Exogenous Retroviruses in OWM Species

Four retrovirus species are currently circulating in OWM, namely SFV, SIV, SRV, and STLV. SFV is also widespread in apes and New World monkeys (NWM). STLV is present in both Asian and African monkey species and in apes plus humans, while SIV is solely found in African primate species (including African apes), and recently, in humans (Table 1).

Several limitations apply when reconstructing the history of exogenous retroviruses from epidemiological data:

- Sampling is not systematic; some species remain untested because they may be difficult to reach.
- Sampling may not be optimal: it is done at the wrong age (for instance SIV is a sexually transmitted infection with juveniles normally being negative for the virus), the wrong type of sample is taken, or the viral load is below the detection level.
- The distinction between an exogenous and endogenous retrovirus is not always clear (e.g., murine leukemia virus, MuLV, feline leukemia virus, FeLV, and koala retrovirus, KoRV, have both infectious and endogenous variants). Moreover, every type of retrovirus has the capacity to enter the mammalian germ line, so Mendelian inheritance is not a distinguishing characteristic [42–44].

#### 3.1.1. Simian Foamy Virus (SFV)

Spumaviruses, also known as foamy viruses due to the effect they induce in cell cultures, are a widely spread, distinct type of retrovirus, which share some similarities with hepadnaviruses with regard to their replication cycle [45]. Infectious spumaviruses are found in OWM, NWM and apes, where they induce little pathology, as replication only takes place in epithelial cells of the oral cavity [45–47]. The detection of endogenous FV-like sequences in a wide variety of amphibians, fish and mammals suggests a considerable history for the group and indeed, phylogenetic analysis infers FVs as the oldest retrovirus lineage, emerging >450 mya [48]. The origin of prosimian FV was calculated to the Mesozoic (~82.5 mya). The divergence between OWM and ape SFV was dated to the Oligocene (~30 mya), with the OWM node emerging ~16 to 18 mya [49]. Species-specific FV clades are more recent; for instance, orangutan genus and subspecies-specific clades have been dated to the late Pliocene (>4.7 mya), and the Pleistocene (>1.7 mya), respectively [50]. Cross-species transmissions and recombination events, however, complicate the reconstruction of spumavirus history [51,52].

**Table 1.** Overview of Old World primate species naturally infected with exogenous retroviruses.

| Retrovirus Species *Genus* | African OWM Species + | | Asian OWM Species + | | Other Primate Species + | References |
|---|---|---|---|---|---|---|
| | **Cercopithecinae** | **Colobinae** | **Cercopithecinae** | **Colobinae** | | |
| Simian foamy virus (SFV) [1] *Simii-spumavirus* | Cercopithecini: *Chlorocebus Erythrocebus* Papionini: *Cercocebus Lophocebus Macaca Mandrillus Papio Theropithecus* | *Colobus Procolobus* | Papionini: *Macaca* | *Pygathrix Trachypithecus* | *Gorilla Hylobates Pan Pongo* | [50,53–56] |
| Simian immuno-deficiency virus (SIV) [2] *Lentivirus* | Cercopithecini: *Allenopithecus Cercopithecus Chlorocebus Miopithecus* Papionini: *Cercocebus Lophocebus Mandrillus* | *Colobus Piliocolobus Procolobus* | None | None | *Gorilla Homo* (HIV) *Pan* | [54,57–62] |
| Simian type D retrovirus [3] (SRV) *Betaretrovirus* | None | None | Papionini: *Macaca* | *Semnopithecus?* | None | [63–67] |
| Simian T-lymphotropic virus (STLV) *Deltavirus* | Cercopithecini: *Allenopithecus Cercopithecus Chlorocebus Erythrocebus Miopithecus* Papionini: *Cercocebus Lophocebus Macaca Mandrillus Papio* | *Piliocolobus Procolobus* | Papionini: *Macaca* | *Presbytis* | *Gorilla Homo* (HTLV) *Hylobates Pan Pongo* | [22,24,54,68–77] |

[1] SFV is also widespread in New World monkeys (NWM) species [51,55,78]. [2] Three natural infections of a yellow, an olive and a chacma baboon with a *Chlorocebus* (African green monkey, AGM) SIV strain, respectively, have been reported [79–81]. SIV isolated from an *Erythrocebus* monkey likely also results from SIVagm cross-species transmission [82]. [3] Simian type D betaretroviruses are recombinants with a betaretroviral *gag-pol* sequence and a gammaretrovirus *env* gene.

An overview of the estimated contemporary distribution of OWM retroviruses and the deduced prehistoric virus migration routes, is given in Figure 1.

3.1.2. Simian Immunodeficiency Virus (SIV)

SIV infection has been detected in African, but not Asian primates or NWM, suggesting that origin and spread of the virus occurred after the separation of OWM and NWM, and also after the major migrations of African primates to Asia in the Miocene period.

Two independent integrations of an endogenous lentivirus, termed pSIV, with a phylogenetic position in between the feline lentivirus feline immunodeficiency virus (FIV) cluster and modern simian SIV strains, have been detected in the genome of Malagasy lemur species, suggesting that SIV precursors infecting prosimian primates already existed ~4.3 mya [83–86]. Since Madagascar and Africa have been separated for at least 160 mya,

transitional pSIV was somehow transmitted across a substantial body of water, indicating that lentivirus dispersal does not always need a land bridge [83].

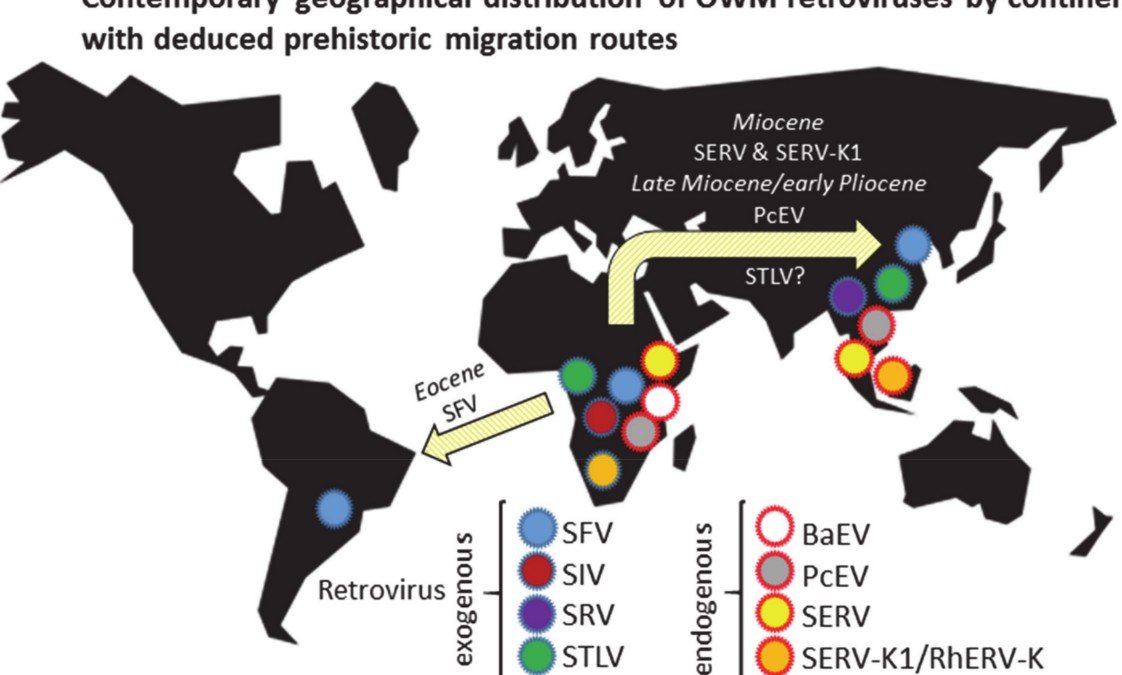

**Figure 1.** Estimated present-day distribution by continent and proposed prehistoric migration routes of eight retroviruses that are, or have been, circulating in Old World monkeys (OWM). Eocene: ~33.9 to 56 mya; Miocene: ~5.3 to 23 mya; Pliocene: ~2.5 to 5.3 mya. The continental positions shown represent the current situation, the time periods shown are estimates. During the Eocene, South America and Africa were much closer to each other. During the Miocene, a junction existed between Africa and Asia due to the closure of the Tethys Seaway.

SIV is remarkably widespread, having infected over 40 African primate species and at least 9 genera (Table 1). Prevalence in populations can be high. For instance, up to 80% of wild *Chlorocebus* females test positive for SIVagm infection [87,88]. SIV strains exhibit relatively large nucleotide diversity, suggestive of great age [87,89]. Initial calculations, however, indicated that SIV lineages evolved in historic times, on a time scale of only tens or hundreds of years [90,91]. Subsequent analysis of SIV isolates from primate populations on the island of Bioko, which is located 32 km off the west coast of Africa and has been separated from mainland Africa for at least 10,000 years, pushed the emergence of present-day SIV back to ≥32,000 years before present [92]. Confusingly, African green monkey (AGM, genus *Chlorocebus*) subspecies each harbor a distinct lineage of SIVagm. This suggests a long-time presence of the virus, with an origin dating to somewhere between AGM subspeciation, estimated at 1.5–3 mya, and subsequent monkey migrations during the Plio-Pleistocene [88]. Should SIV indeed have a relatively short history in African primates, then it is understandable that the virus did not reach Asian monkey species. The Plio-Pleistocene is known for its lower global temperatures and, in the Pleistocene, repeated glaciation. Such climate conditions discouraged primate settlements in Europe and Arabia that could bridge the gap between Africa and Asia [93]. *Theropithecus* species did disperse to Europe in the early Pleistocene [94]. However, modern *Theropithecus* is not infected by SIV, nor are baboons, which currently still inhabit Saudi Arabia and Yemen. *Papio hamadryas* likely migrated from Africa to Arabia in the late Pleistocene and remained there ever since [95,96]. Looking at the spread of SIV, it is commonly found in arboreal primates inhabiting tropical forests rather than terrestrial, savannah or shrubland-dwelling monkeys such as macaques and baboons (Table 1). The exception to this observation are

green monkeys. As a widespread species in sub-Saharan Africa, where they can be found in numerous environments, AGMs may function as an intermediary spreading SIV to savannah monkeys. It is interesting to note that at three occasions, wild baboons were found to be infected with SIVagm strains [79–81]. A Senegalese patas monkey (*Erythrocebus patas*), another ground-dwelling species living among AGMs, was likewise reported to carry a SIVagm strain [82].

Concluding, a relatively young age has been calculated for SIV based on nucleotide sequences and from geographical separation. Such a young age is supported by its absence in Asian primates, suggesting a post-Miocene origin. Probably, the virus originates from the tropical forests of sub-Saharan Africa and is now spreading to ground-dwelling primates through AGMs, which are widespread in Africa, have high SIV prevalence, and thrive in many habitats. AGMs themselves have probably obtained SIV from cross-species transmission to one of their subspecies with subsequent onwards transmission to the other AGM subspecies in the past [97].

### 3.1.3. Simian Type D retrovirus (SRV)

SRV, a primate betaretrovirus, which can induce considerable pathology in captive macaques, was found to have natural reservoirs in the Asian macaque species *Macaca nemestrina*, *Macaca fascicularis*, and *Macaca mulatta* [63,64,66,98]. In total, eight phylogenetically related SRV genotypes have been identified, all in macaques except SRV-6. This variant was detected in wild Hanuman langurs, *Semnopithecus entellus*, from India [67], suggesting that SRV-like viruses could have a broader species distribution. However, it has been noted that SRV is closely related to endogenous primate type D betaretrovirus SERV sequences. Especially the single SRV-7 sequence available, a partial fragment of *pol*, shows high homology to SERV *pol* from baboons (unpublished observation, and see [66,99]). Jayashree Nandi et al. therefore suggested that SRV-6 and -7 may be the result of recombination events between exogenous SRV and endogenous SERV genomes [66], which would explain their considerable genetic distance from SRV 1–5 and SRV-8. As SRV-6 also bears resemblance to an endogenous type D retrovirus (PO-1-Lu from *Trachypithecus obscurus*, a langur species from East Asia), it could also be a reactivation of such a provirus [100]. PO-1-Lu was shown to be very similar to Mason–Pfizer monkey virus (MPMV) isolated from a rhesus macaque, which turned out to be a variant of SRV-3 [101,102]. These difficulties show that type D endogenous and exogenous retroviral sequences are intertwined and that recombination, reactivation and/or germ line integration has probably happened several times, which greatly confuses SRV phylogeny.

Little is known about the origin and age of SRV, except for its spread in primate centers since at least the 1970s [99,103]. SRV 1–5 and SRV-8, which are genetically and serologically closely related, spread only within the genus *Macaca*, and have the ability to induce severe disease. Such characteristics are typical of a recently evolved virus species, making the putative recombinant SRV-types 6 and 7 detected in Indian langurs a distinct viral species. Increasing the search for both infectious and endogenous SRV-like retroviruses in Asian monkeys should help gain a better understanding of the distribution and evolutionary connections within this virus family.

### 3.1.4. Simian T-Lymphotropic Virus (STLV)

The deltaretrovirus STLV comprises multiple types and subtypes and has a broad distribution in OWM and apes, but is not found in NWM [104]. Phylogenetic analysis of LTR sequences suggests an African, rather than Asian origin for PTLV (primate T-lymphotropic virus, the collective name for STLV and HTLV, the *Homo*-specific variants that arose after cross-species transmission) [105,106]. Three STLV types have been recognized, all of which also infect humans [107]. HTLV-1 and HTLV-2 have by now been scattered around the world by human migration, while HTLV-3 is only found in Africa [107]. A fourth, distantly related type, HTLV-4, has been isolated from an individual in Cameroon, but the search for a simian counterpart has so far been unsuccessful [108].

Pathogenicity in humans and non-human primates has only convincingly been documented for PTLV-1 (the cluster to which both STLV-1 and HTLV-1 belong, reviewed in [76,109]). PTLV mostly replicates through clonal expansion, which results in a low mutation rate and a high genetic stability [110]. However, diversity is significant in this large virus family, making phylogenetic analysis of PTLV challenging. Often, studies are based on relatively limited information. For instance, transmission of STLV-1 to humans was estimated to have occurred 27,300 ± 8200 years ago based only on the analysis of LTR and *env* gene sequences [21]. However, further transmissions have taken place over time or are still ongoing (discussed in [76]), whereby geographic proximity is the basis of most shared evolutionary relationships [23]. In fact, each HTLV-1 subtype is probably the result of independent transmissions from either African or Asian simians [24,111]. Analysis using a relaxed molecular clock and only first and second codon positions of the *tax* and *env* genes, respectively, suggested that HTLV-4 diverged from HTLV-2/STLV-2 approximately 49,800–378,000 years ago [28]. In this study, the most recent common ancestor (MRCA) of the major PTLV clades was calculated to have existed between 214,650 (*tax* gene) and 385,100 (*env* gene) years ago [28]. Using amino acid sequences from 20 PTLV strains and a bovine deltaretrovirus as outgroup, another study suggested a much older date of origin for the PTLV MRCA, namely, >1.3 mya [26]. A third study, based on entire genomes without LTRs, suggested an age between 632,129 and 946,936 years for the whole PTLV cluster, and dated the subclades of PTLV-1, PTLV-2, and PTLV-3 to 53,000–79,684, 191,621–286,730, and 63,294–94,770 years ago, respectively [112].

Overall, it is clear that PTLV comprises a clade of ancient, genetically diverse viruses, where repeated interspecies transmissions as well as viruses migrating between Asia and Africa pose a challenge for those trying to reconstruct the history of the virus. Despite some variation in estimated divergence dates, the above studies all agree on a late Pleistocene origin for PTLV, with the three main types likely arising in the last 200,000 years.

### 3.2. Endogenous Retrovirus Integrations Predating OWM Speciation

Endogenous retrovirus genomes and parts of such proviruses abound in vertebrates. Most have entered the germ line long before the rise of the order of primates. The primate germ line was thus already seeded with many of such proviral remains before the diversification of extant OWM species. Examples of ancient proviruses, with their calculated time of integration, are human endogenous retrovirus-H (HERV-H, >40 mya), HERV-W (±63 mya), HERV-S (±43 mya), HERV-R (±33 mya), HERV-I (±33 mya), and HERV-E (10.7–41.3 mya) [113–118]. With the exception of HERV-S, which has some similarity to foamy viruses, all these HERVs belong to the gammaretrovirus group.

The situation for the supergroup of primate betaretroviruses, represented by HERV-K viruses in humans, and CERV-K in chimpanzees, is more complex [119,120]. Sequences homologous to HERV-K have been detected in all Old World primate species, with an estimated integration time of 28 mya [121]. However, specific members of the family show a more limited distribution and a later germ line introduction, suggestive of repeated activation and reinfection processes over time (discussed in Section 3.3.4).

### 3.3. Endogenous Retroviruses Specific for OWM

Four endogenous retroviruses are exclusively present in OWM genomes, namely, BaEV, PcEV, SERV, and SERV-K1/RhERV-K. Of these viruses, full-length proviral genomes can be detected in at least some OWM species [37–39,122,123]. In AGM (Vero) or baboon BEF-3 cell lines, virus particles containing SERV or BaEV genomes, respectively, can be produced upon stimulation [124,125]. For PcEV, expression has only been confirmed at the RNA level [126]. It is unknown whether this RNA comprises full-length genomes nor whether it is packaged. Similarly, for SERV-K1, only *env* mRNA expression has been verified [123]. Distribution of the four endogenous retroviruses among OWM species is shown in Table 2; corresponding geography and deduced virus migrations are depicted in Figure 1.

Several limitations apply when reconstructing the history of endogenous retroviruses from genome assemblies:

- Not all infected species may still roam the earth
- Not all infected species may contain germ line integrations
- Not all species have had their genomes sequenced
- Not all proviral sequences are likely due to bona fide viral infection and integration; they may for instance be acquired by hybridization between species
- Proviruses may have been completely or partially lost from the germ line, making identification challenging
- The quality of a genome assembly may be insufficient for provirus detection
- Heterozygous proviral insertions (ERV insertion polymorphism) are not included in genome assemblies [127,128]
- The distinction between endogenous and exogenous retroviruses is not always clear (see above, and the comment in [129])

**Table 2.** Overview of OWM species harboring OWM-specific endogenous retrovirus genomes.

| Retrovirus Species *Genus* | African OWM Species + | | Asian OWM Species + | | References |
|---|---|---|---|---|---|
| | Cercopithecinae | Colobinae | Cercopithecinae | Colobinae | |
| Baboon endogenous virus (BaEV) *Gammaretrovirus* | Cercopithecini: *Chlorocebus* Papionini: *Cercocebus* *Mandrillus* *Lophocebus* *Papio* *Theropithecus* | None | None | None | [130] Figure A1 |
| Papio cynocephalus endogous virus (PcEV) *Gammaretrovirus* | Papionini: *Lophocebus* *Papio* *Theropithecus* | *Colobus* | Papionini: *Macaca* | None | [131] Figure A1 |
| Simian endogenous retrovirus (SERV) *Betaretrovirus* *(Simian type D)* | Cercopithecini: *Cercopithecus* *Chlorocebus* *Erythrocebus* *Miopithecus* Papionini: *Macaca* | None | Papionini: *Macaca* | *Rhinopithecus* | [38,132,133] Figure A1 |
| Simian endogenous retrovirus-K1 (SERV-K1/ RhERV-K) *Betaretrovirus* | Papionini: *Cercocebus* *Mandrillus* *Papio* *Theropithecus* | *Colobus* | Papionini: *Macaca* | *Trachypithecus* | Figure A2 |

### 3.3.1. Baboon Endogenous Virus (BaEV)

BaEV was discovered as an integrated provirus in baboon DNA in 1978 and was completely sequenced in 1987 [39,134]. Non-infectious virus particles containing BaEV sequences can be induced from the Vero cell line, which is derived from *Chlorocebus* kidney epithelial tissue [124]. BaEV is a recombinant virus, originating from the now also endogenous monkey retroviruses PcEV and SERV [37]. Most BaEV proviral integrations are defective, although complete proviruses do exist in baboons and geladas (Appendix A Figure A1A) [122,125,134]. Although BaEV was at first thought to be widespread in primates, a PCR analysis of 24 African monkey species suggested that BaEV integrations are limited to few species (Table 2) [130]. In phylogenetic reconstructions, BaEV sequences did

not follow the monkey phylogeny, but clustered according to habitat, implying that living together facilitates cross-species transmissions [130]. The BaEV_forest and BaEV_savannah strains were estimated to have diverged 24,000–400,000 years ago, using evolutionary rates calculated for other retroviruses [130]. Analyzing LTR divergence in full-length BaEV genomes from one baboon and two *Theropithecus gelada* showed a single (baboon) or no (gelada) substitutions between the 5′ and 3′ LTR, respectively (unpublished observation). As retroviral LTRs are identical at the time of integration, after which they start to diverge with the host mutation rate, little or no LTR variation is indicative of a relatively recent origin, probably at most ~300,000 years ago for proviruses with identical LTRs [135].

### 3.3.2. Papio Cynocephalus Endogenous Virus (PcEV)

The full-length endogenous gammaretrovirus PcEV was found by screening a baboon (*Papio cynocephalus*) genomic library [37]. Additional species harboring PcEV proviral sequences were identified during the analysis of OWM monkey DNA (Table 2, [131]). Interestingly, PcEV sequences (including LTRs) were found in all examined papionin species, but also in colobines of the genus *Colobus*: *Colobus guereza* and *Colobus angolensis* (Appendix A Figure A1B, [131]). PcEV_baboon was not found in Cercopithecini, although a variant virus is likely present there [131]. The upper limit of PcEV integration was set at 9 mya based on LTR divergence and species distribution, such as the dispersal of macaques into Asia around 5.5–7 mya [131]. Evidence for recent activity is given by proviruses with no or only a single nucleotide difference between their 5′ and 3′ LTRs, which can be identified in gelada and baboon genomic libraries, respectively (unpublished observation). In the rhesus macaque genome, PcEV was identified as one of the retroviruses with relatively recent germ line activity [136] (see erratum in [126] as PcEV was mistakenly labelled BaEV in the earlier publication). PcEV integration sites in two olive baboons (no. 15944 and no. 1X1155) differ, suggesting that the proviruses are not fixed in this subspecies (unpublished observation). In addition, baboon PcEV integration sites are empty in the rhesus macaque genome, which points to independent integrations after speciation and not inheritance from a common ancestor (unpublished observation).

So, as PcEV is present in *Colobus*, but not in Cercopithecini, the exogenous period of PcEV can be estimated to have started around the macaque/papionin split, 7.6 ± 1.3 mya, until about 150,000–300,000 years ago. An ancient cross-species transmission to an ancestor of present-day *Colobus* species could explain its presence there [131].

### 3.3.3. Simian Endogenous Retrovirus (SERV)

Similar to PcEV, SERV, a full-length integrated type D betaretrovirus was first described from a baboon genomic library [38]. Subsequently, a widespread distribution in OWM with a relatively large nucleotide divergence between the viral sequences suggested a relative old age for the proviruses (Table 2) [38,132]. Heterozygous integrations abound, at least in *Chlorocebus sabaeus*, which suggests that many are so recent that they have not had sufficient time to become fixed in the population [128]. Fixation of traits is highly dependent upon effective population size, which depends on population size and generation time. For primate populations 1–3 million years are usually sufficient to fix most markers [137], suggesting that polymorphic insertions are less than 1–3 my old. A number of OWM genomes contain several full-length SERV genomes, often with open reading frames for at least some viral proteins (unpublished observation, Appendix A Figure A1C). SERV sequences do not cluster according to host species, implying that they integrated after speciation and that ancient cross-species transmissions occurred [38]. SERV particles can be expressed from Vero cells, but those are non-infectious in cell lines known to be permissive for type D retroviruses [124].

Full-length proviruses, intact reading frames, virus expression and heterozygous insertions do suggest a young age, but SERV has been calculated to have integrated on average 6.16 ± 3.41 (range 0-21.62) mya in Asian colobines, which, however, carry a distinct variant of SERV, and somewhat later, 3.42 ± 2.20 (range 0.27–14.09) mya, in cercopithecine

species; that is, monkeys of the subfamily Cercopithecinae; thus, both Cercopithecini and Papionini [132]. It was therefore suggested that SERV originated within the last 8 million years and continued its exogenous activity until quite recently. For instance, in Asian colobines, species specific, genus specific, but also shared integrations are seen [132]. Highly divergent SERV 5′ and 3′ LTRs can be found in cercopithecines. In contrast, in *Theropithecus* and *Chlorocebus* there are also a few proviruses with only 1–2 nt substitutions between the LTRs (unpublished observation), indeed pointing to a lengthy period of germ line activity. *Rhinopithecus roxellana* SERV integrations with open reading frames were found to be significantly younger than the ones with frame-shift mutations [132]. Furthermore, heterozygous SERV insertions are presumed to be more recent than integrations that have been fixed in a population.

### 3.3.4. Simian Endogenous Retrovirus-K1 (SERV-K1)/RhERV-K

The primate non-D type betaretrovirus superfamily, to which all ERV-K viruses belong, has a complex history. Deep-rooted proviral integrations are present in all OWM and hominoids, but there is also evidence of later activity. In both humans and chimpanzees, the possession of open reading frames combined with low LTR diversity in some proviruses as well as the existence of polymorphic insertions, are illustrative of such recent activity [127]. In case of chimpanzee CERV-K proviruses, integration times postdate the *Homo–Pan* divergence [127]. Another study dated the average integration time of rhesus macaque ERV-K (RhERV-K) at 10.3 mya, although almost identical LTRs in three complete proviruses suggested that some integrated <4.5 mya [120]. SERV-K1, a full-length ERV-K-like provirus, has been described in the rhesus macaque [123]. Most likely, SERV-K1 is identical to one of the RhERV-K integrations, but unfortunately RhERV-K sequences are no longer available in the database.

Southern blot analysis indicated that ERV-K proviruses, HERV-K(OLD), are present in all OWM [121]. However, as more recent, lineage specific ERV-K insertions have been reported, the presence of such ERV-K proviruses in OWM was investigated by querying their genome assemblies with full-length HERV-K and SERV-K1 sequences. Although proviral fragments with substantial homology (79–96%) were indeed detected in all OWM species, phylogenetic analysis as well as their chromosomal locations suggested that these ERV-K-like sequences had been present before the radiation of OWMs, and thus represent HERV-K(OLD) insertions [121]. In contrast, full-length or near full-length proviruses were found in macaques, in the gelada, and in the olive baboon genome assemblies when using SERV-K1 as query (Table 2, Appendix B Figure A2). The coding regions of the SERV-K1-like proviruses were highly homologous (80–99% homology over substantial genomic lengths) to the HERV-K family, including its ERV-K(OLD) members. Fortunately, SERV-K1 LTRs proved to be relatively unique and suitable for virus identification. However, such acquisition or exchange of (novel) sequence stretches, which appears to be quite common for the ERV-K family, makes tracing its history through the primate germ line a challenging undertaking. It is thus not easy to identify the species infected with those specific SERV-K1 viruses, as only *Macaca mulatta* and *Macaca fascicularis* proviruses did contain both LTRs, while those in *Macaca fuscata* lack them. Furthermore, SERV-K1 proviruses are incomplete in *Macaca nemestrina*, although the particular LTR sequences are present (unpublished observation). Asian *Trachypithecus francoisi* and African *Colobus angolensis*, but not other colobine species, do contain integrations with LTR sequences that are largely homologous to SERV-K1 LTRs. Considering its age and spread, SERV-K1 may have migrated with its colobine and papionin hosts to Asia in the late Miocene. Comparing SERV-K1 LTR sequences from rhesus macaque full-length proviruses shows that the 5′ and 3′ LTRs have no, or very low, nucleotide diversity (unpublished observation, Appendix B Figure A2), suggestive of relatively recent integration events. Apparently, SERV-K1 activity started before the primate Out-of-Africa migrations and continued for quite some time after that. The occurrence of a few very similar SERV-K1 proviruses in rhesus and crab-eating macaque genomes (unpublished observation) can likely be explained by interbreeding [138]. Several

SERV-K1 LTRs in papionins are almost identical in sequence and could thus predate speciation. A quick inspection of Blast hits, however, suggests that the integration sites do not match, so that the proviral integrations must largely have been independent events (result not shown).

The uneven distribution of SERV-K1 complete proviruses suggests that either the infectious virus did not always reach the OWM germ line, that the integrations were lost, or that they are somehow not in the genome assemblies. Overall, due its apparently highly infectious nature over long periods of time, and a tendency to recombine, the epidemiological trail of the ERV-K virus family is not easy to follow. The recent spread, on two continents, of two related SERV-K viruses in two very different monkey genera, papionins and *Colobus/Trachypithecus*, is thus an interesting subject for further study.

Infectious period estimates of the eight OWM retroviruses in relation to the geological time scale are summarized in Figure 2.

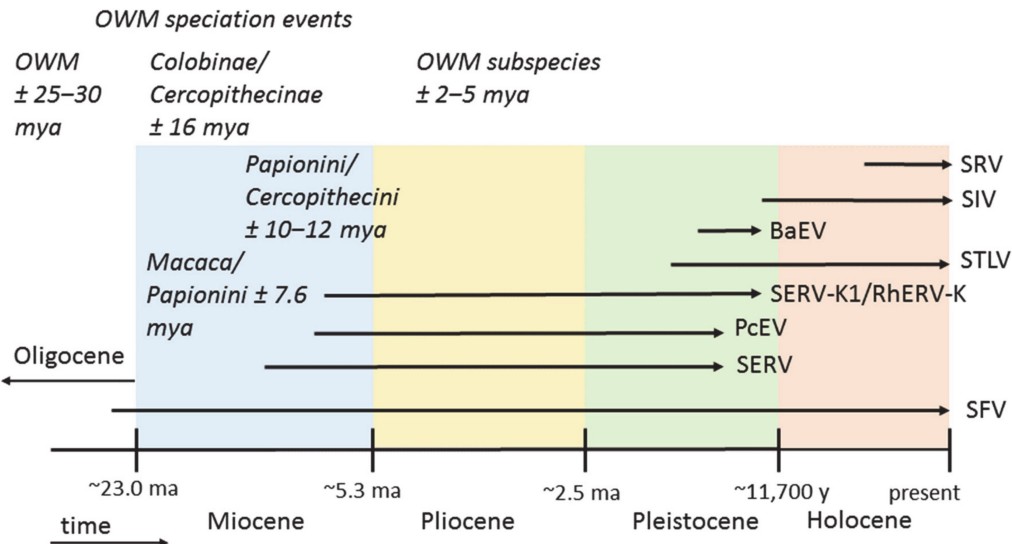

**Figure 2.** Estimated infectious periods of eight retroviruses that are, or have been, circulating in OWM. Putative speciation events in OWM are indicated. The geological time line is not drawn to scale.

## 4. Discussion

The order of primates probably dates to the Paleocene (56–65 mya). Primate fossils appeared simultaneously in Europe, Asia, and Africa in the early Eocene (55–50 mya), suggesting that by then, the order was firmly established. Since primates are relatively intolerant of cooler climates, their history is largely determined by climatic variations, which in their turn are linked to tectonic and cosmic events [93]. The rapid evolution and dispersal of primate species was likely facilitated by a steep rise in global temperature at the Paleocene/Eocene boundary, which resulted in tropical climates even at higher latitudes [93]. Moreover, firm geographical connections between Asia, Europe, and North America existed at that time, giving the early primates sufficient territory to expand. Global temperatures declined during the Eocene, although warmer periods, for instance in the Miocene, would continue to occur. The much colder conditions in the Oligocene and especially in the more recent Pliocene and Pleistocene periods forced primates to retreat from previously inhabited regions. Under dryer, colder conditions, terrestrial papionins, like macaques and baboons, could from time to time establish themselves in such climates, but the arboreal monkeys of the tropical forests were unable to adapt. For instance, colobines migrated successfully from Africa to Eurasia in the late Miocene but these dispersals stopped in the Plio/Pleistocene; *Papio* and *Theropithecus* were the only

species then trekking to Eurasia through Arabia [139]. Macaques migrated from Africa to Eurasia around 5.5–7 mya in the late Miocene or early Pleistocene [138]. Only a single, ancestral species, *Macaca sylvanus,* is now left in (northern) Africa; European macaques have gone extinct in the late Pleistocene [140]. Modern Asian macaque species show no genomic evidence of later introductions, suggesting that subsequent migrations, if any, have been unsuccessful [138].

With the above described primate evolution in mind, an attempt to retrace the history of one of their successful parasites, the retrovirus family, was made. Retroviruses are generally transmitted by close contact, for instance by the exchange of saliva or blood (biting), through sexual encounters, or from mother-to child, implying that these pathogens are closely associated with their host species. Infections by aerosols, the oral–fecal route or by insect vectors are relatively rare or are unlikely; only equine infectious anemia virus, EIAV, and possibly bovine leukemia virus, BLV, can sometimes be transmitted by flies from viremic hosts [141,142]. So, in general, the history of a retrovirus is closely related to that of its host. Of course, transmission of viruses can be hampered by cellular and genetic variation between species. For SIV, immune system activity, including the actions of restriction factors such as TRIM5alpha and APOBEC3G, as well as amino acid variation in for instance the coreceptor protein, have all been implicated in the prevention of cross-species transmission (for a review, see [143]). Such inhibitory mechanisms could lead to false conclusions as to why certain species are free of infection but would hardly play a role in conclusions drawn from infected species.

Investigating the spread and estimated age of eight OWM-infecting retroviral species suggests that SFV, the most widely spread virus of all, with a distribution over three continents, is also the oldest virus in the group. Ancestral primates most likely migrated, possibly by rafting, to South America during the Paleogene when that continent was relatively close to Africa [93]. As the nodal ancestor of OWM and NWM SFV has been dated to ~65 mya [49], the virus could have arrived in South America together with its host. A nodal ancestor by definition predates its progeny, and need not have been a primate virus, which would date the emergence of genuine SFV-NWM and SFV-OWM after 65 mya. Various approaches have estimated the node separating NWM and OWM at ~40 mya in the Eocene, which would fit with the ancestry of the virus lineages [2]. However, as cross-species transmission and recombination have been documented for foamy viruses, alternative scenarios with a later introduction in NWM could also be a possibility.

The next oldest viruses, SERV and SERV-K1 (RhERV-K), are now known only in an endogenous form. Their putative Miocene origin, together with a wide distribution in OWM species in both Africa and Asia, suggests that SERV and SERV-K1 have been travelling the land bridge(s) between the continents established after the final closure of the Tethys Seaway ~14 mya [144]. The relatively warm climate of the Miocene facilitated the dispersal of arboreal, tropical colobines as well as terrestrial macaques; the genera underwent an extensive diversifying radiation in Asia [139]. A third virus with a similar distribution, both in species and geography, is STLV. However, phylogenetic analysis indicated that the common ancestor of STLV genotypes dates to ~1 mya, so that spread of STLV cannot readily be explained by Miocene migrations. A later introduction with rapid onward transmissions and adaptation could be hypothesized, though PTLV is mainly transmitted through sexual contact, blood, or saliva exposure (biting), or vertically from mother to child, which makes spread in and between primate groups relatively slow [145,146]. Alternatively, STLV might be much older than estimated from sequence data, as we have possibly not yet sampled all PTLV variation, or the evolutionary models used to analyse PTLV may be incorrect [76,147].

Another virus that likely migrated with its host from Africa to Asia is PcEV, which is found in African papionins as well as in Asian macaques. Such a distribution suggests that PcEV is at least 5.5–7 mya old, but not much older as the virus is not found in the germ line of the Cercopithecini, implying that migration with its macaque host took place in the late Miocene or early Pliocene. Recombination of PcEV with SERV, when at least one of the two viruses was actively replicating, gave rise to BaEV, which, by definition, should

be younger than its parents. Indeed, BaEV is not found in Asian macaques, suggesting its infectious period postdates the divergence of macaques from the other papionins. The putative Pleistocene origin of BaEV prohibited its spread to Asia, as by then the climate was no longer favorable for monkey expansions. BaEV may have been present in European baboons, having gone extinct there together with its host. Moreover, it would be interesting to investigate whether or not African *Macaca sylvanus* carry PcEV and/or BaEV integrations.

Much younger OWM retroviruses appear to be SIV and SRV; SIV has an exclusively African distribution while SRV is seemingly only found in Asian macaques. The uninhabitable nature after the Miocene period of areas in key corridors for primates between Africa and Asia, with subsequent isolation of monkey species on both continents, likely impeded virus transfer between continents for these recently emerged viruses [148–152]. For a relatively young virus that owes its present global spread to a recent leap to mankind, SIV has a remarkably broad distribution in African primates. Unexpectedly, the list of SIV-infected species shows only anecdotal detection in terrestrial papionins such as baboons and geladas, while infections are common in monkeys and apes inhabiting the tropical forests of sub-Saharan Africa, including the forest-dwelling papionins *Mandrillus*, *Cercocebus,* and *Lophocebus*. An exception here are AGMs (*Chlorocebus* spp.), found just about everywhere in sub-Saharan Africa, which carry a great diversity of SIV strains and have high SIV prevalence. They may have facilitated SIV cross-species transmissions in the past, similar to modern AGM transmitting SIVagm to individual baboons and patas monkeys. An example of another retrovirus shared between AGMs and papionins is BaEV.

For endogenous viruses that have gone extinct, once infected species may be missed because no germ line introductions occurred or were lost in subsequent generations. Still, the available data on endogenous viruses can be used to infer a plausible evolutionary trajectory. Retroviruses that use the neutral amino acid transporter ASCT2 as a cellular receptor, which is expressed on oocytes, spermatozoa, and pre-implantation embryos, have a tendency to enter the germ line [153–155]. Such viruses, here represented by the type D viruses SERV and BaEV, are thus well equipped to create a broad genomic track, and will therefore be less likely to disappear from sight. Indeed, SERV has an extensive distribution in OWM, while—putative—non-ASCT2 using PcEV has a relatively limited distribution in papionin genomes only. However, although BaEV is able to the ASCT2 receptor, it has a more limited distribution than SERV, which may either be a true representation of the species infected in the past but could also be due to other causes. For instance, BaEV can use the neutral amino acid transporter ASCT1 as an additional receptor, which other type D viruses cannot [156]. Such variation in receptor use may suggest an alternative trajectory in the host, although ASCT1 has a similar broad tissue expression as ASCT2 [157]. Alternatively, superinfection resistance (SIR), whereby for instance a homologous Env protein occupies the viral receptor so that new infections cannot take place, could play a role here (for a review, see [158]). The expression of endogenous SERV Env would make such a scenario possible. In human placenta, a HERV-F Env protein has been shown to inhibit betaretroviral Env protein mediated cell fusion through ASCT2 binding [159].

Interestingly, all four now endogenous OWM retroviruses have disappeared from the circulation, suggesting that virus endogenization ultimately means the end of the contagious variant. SIR could be such a possible mechanism to limit virus spread. Thus, opting for germ line transmission seems to be an indicator for extinction of infectious retrovirus lineages, while old age, as exemplified by the current infectivity of SFV, does not need to be.

## 5. Conclusions

Current distribution of primates, prehistoric primate evolution and expansion together with virus prevalence and sequence information have been used to infer the history of primate retroviruses. African and Asian monkey species are, or have been, infected by at least eight distinct retroviral species. Four of those have now become part of the germ line while four others are currently in the infectious stage, including the most ancient retrovirus,

SFV. Evidence suggests that the older retroviruses—PcEV, SERV, SERV-K1—have migrated with their hosts to Asia ~5 to 10 mya, while the more recent ones—BaEV, SIV, and SRV—are limited to their respective continents of origin. For STLV, the reconstruction of its history proved more difficult, as its relatively recent age and widespread distribution does not correspond with Miocene migration.

**Supplementary Materials:** The following are available online at https://www.mdpi.com/2673-3986/2/1/5/s1, File S1: Alignment of OWM ERV sequences, File S2: Alignment of OWM SERV-K LTR sequences.

**Funding:** This research received no external funding.

**Institutional Review Board Statement:** Not applicable.

**Informed Consent Statement:** Not applicable.

**Data Availability Statement:** The data presented in this study are available in Supplementary Files 1 and 2.

**Conflicts of Interest:** The author declares no conflict of interest.

## Appendix A

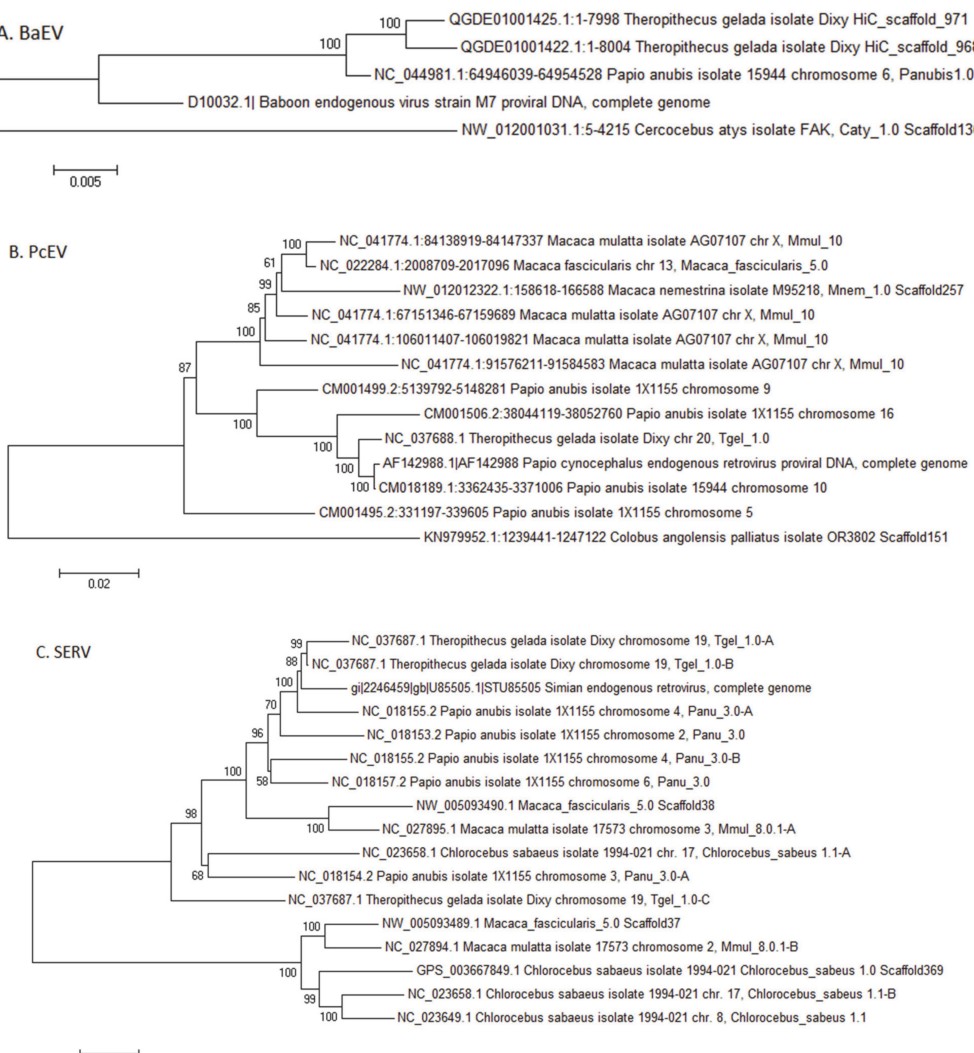

**Figure A1.** Evolutionary relationships of full-length or near full-length proviruses of BaEV (**A**), PcEV (**B**), and SERV (**C**) in OWM.

The evolutionary history of integrated proviruses of BaEV (panel A), PcEV (panel B) and SERV (panel C) in OWM genomes with the corresponding reference virus sequence was inferred using the Neighbor-Joining method [160]. Bootstrap values of 500 replicates are shown next to the branches [161]. The tree is drawn to scale, with branch lengths in the same units as those of the evolutionary distances used to infer the phylogenetic tree. The evolutionary distances were computed using the Kimura 2-parameter method [40] and are given as the number of base substitutions per site. There were a total of 8492 positions in the dataset for BaEV, 8282 for PcEV, and 8382 for SERV. Evolutionary analyses were conducted in MEGA6 [41]. For some species, such as *Macaca nemestrina*, *Mandrillus leucophaeus*, and *Cercocebus atys*, the assembled genome sequences were of relative low quality, in other instances it was ascertained that all viral elements, *gag-pol-env* coding regions and LTRs, were present with high similarity (>95%), although the proviruses themselves were likely fragmented in those species and full-length or near full-length proviruses could not be retrieved. The primate species name is embedded within the sequence name. The sequence alignment file is available as Supplementary File 1.

**Appendix B**

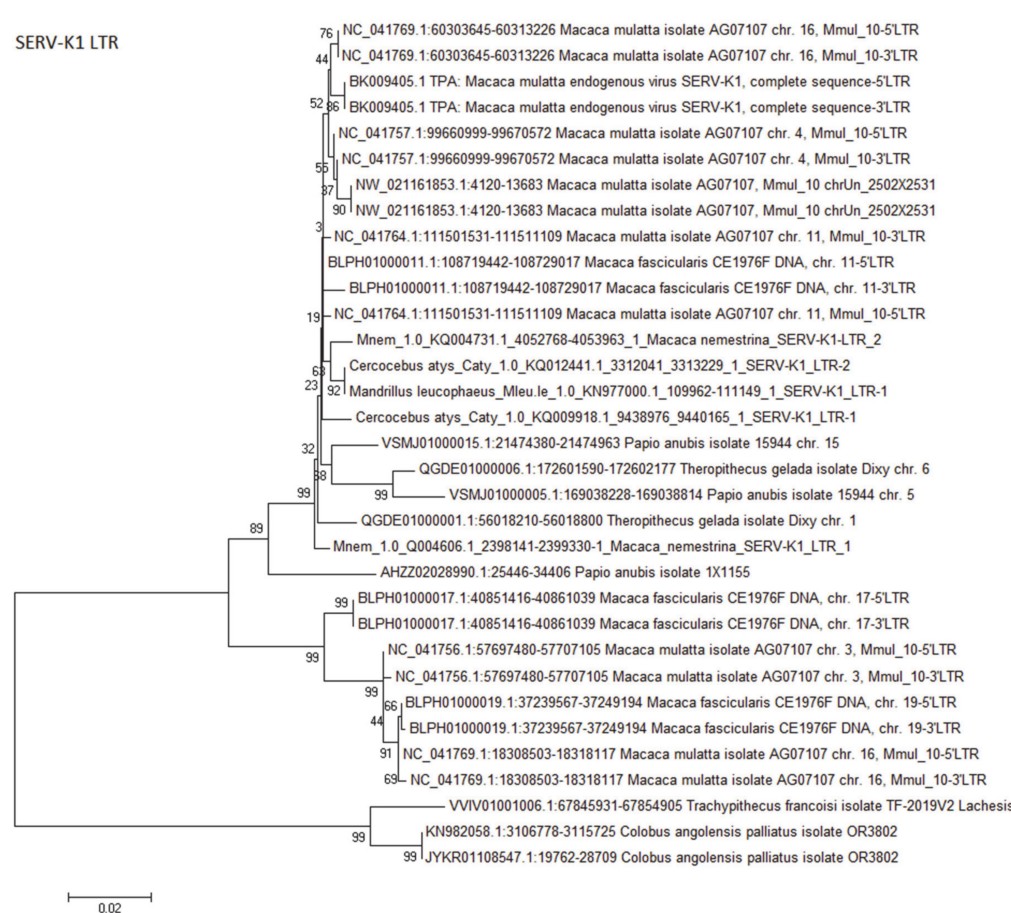

**Figure A2.** Evolutionary relationships of SERV-K1 LTR sequences in OWM.

The evolutionary history of LTR sequences of SERV-K1 in OWM genomes and those of the rhesus macaque reference virus were inferred using the Neighbor-Joining method [160]. For complete proviruses, both the 5′ LTR and 3′ LTR were included in the analysis. Bootstrap values of 500 replicates are shown next to the branches [161]. The tree is drawn to scale, with branch lengths in the same units as those of the evolutionary distances used to infer the phylogenetic tree. The evolutionary distances were computed using the Kimura 2-parameter method [40]. Evolutionary analyses were conducted in MEGA6 [41]. There

were a total of 572 positions in the dataset. The primate species name is embedded within the sequence name. The sequence alignment file is available as Supplementary File 2.

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
