# Peer review of "Contemporary Distribution, Estimated Age, and Prehistoric Migrations of Old World Monkey Retroviruses"

_epidemiologia, doi:10.3390/epidemiologia2010005_

Round 1

Reviewer 1 Report

This is a very interesting paper from an expert on primate retroviruses that combines the best aspects of a comprehensive review and a research article. The manuscript provides a much needed detailed review of the literature on exogenous and endogenous retroviruses of primates and primate genomes, respectively, and incorporates updated bioinformatic analyses by the author. A specific strength of the paper is the delineation of caveats (e.g. Section 3.1) that define the limits of current research and will help prevent confusion on the part of the reader, particularly those readers unfamiliar with endogenous retroviruses. Similarly, the author is commended for pointing out that there is a “gray area” between “exogenous” and “endogenous” and that there is not necessarily a clean separation between the two.

Abstract, first line - author might want to consider writing “are currently affected by “at least” four infectious retroviruses”. Alternatively, could say “currently known to be affected by”. 

There is one significant issue that requires correction: the author mistakenly describes CERV-1 and CERV-2 as part of the ERV-K family (Materials and Methods, 2.2, first paragraph). To the best of my knowledge, these are gammaretrovirus-like ERVs, and are unrelated to the well-studied betaretrovirus-related ERV-K family. Later, in section 3.2 second paragraph, the author also includes PtERV, CERV-1 and CERV-2 as part of the “supergroup of primate betaretroviruses” but taxonomically these would be classified as gammaretroviruses - a fact supported by one of the citations provided by the author (reference 112). PtERV/CERV-1 and CERV-2 are not a major thrust of the review, so this error will only require some minor rewriting of the relevant passages.

In the section on the SIVs, it is surprising that the author does not refer to the prosimian endogenous SIV of lemurs in the discussion of the age of the SIVs. It would seem that the phylogenetic relationship here would also contribute to understanding the time and origin of modern SIVs. It is also more evidence that retroviruses could migrate between bodies of land separated by water (ie a land bridge may not always be required).

It could be clearer that SERV-K1/RhERV-K are phylogenetically part of the larger ERV-K or HERV-K (HML-2) clade, and that it has been well established that the oldest ERV-K proviruses in the human genome have orthologues in OWM (see ref 110).

Section 3.2 “Sequences homologous to HERV-K have been detected in all Old World primate species, with an estimated integration time of 28 mya” - The reference listed here is to a koala endogenous retrovirus paper that does not have any data regarding HERV-K.

Section 3.2.2 “So, as PcEV is present in Colobus, but not in cercopithecine monkeys…” The wording may be confusing to readers not familiar with primate taxonomy and nomenclature - the similarities of Cercopithecinidae, Cercopithecinae, Cercopithecini and in particular the informal term cercopithecine. Clarification of the informal term somewhere in the text would help the reader understand the argument being presented.

Page 6, last paragraph - Macaca should be italicized.

Page 7, the use of the term PTLV precedes the definition of the term.

The reference to PTLV-1 is somewhat confusing - is this a mistake, is it equivalent to STLV-1, or is it a collective term for HTLV-1 + STLV-1? Some clarification is needed here.

Pages 7 and 8 - the viral genes “tax” and “env” should be italicized.

Page 8, near the top: “deltavirus” should be “deltaretrovirus”.

Author Response

Response to Reviewer 1 Comments

First, I am sorry that the template of the manuscript I have to submit for revision again does not include line numbers.

Abstract, first line - author might want to consider writing “are currently affected by “at least” four infectious retroviruses”. Alternatively, could say “currently known to be affected by”. 

Response: changed into ‘at least’

There is one significant issue that requires correction: the author mistakenly describes CERV-1 and CERV-2 as part of the ERV-K family (Materials and Methods, 2.2, first paragraph). To the best of my knowledge, these are gammaretrovirus-like ERVs, and are unrelated to the well-studied betaretrovirus-related ERV-K family. Later, in section 3.2 second paragraph, the author also includes PtERV, CERV-1 and CERV-2 as part of the “supergroup of primate betaretroviruses” but taxonomically these would be classified as gammaretroviruses - a fact supported by one of the citations provided by the author (reference 112). PtERV/CERV-1 and CERV-2 are not a major thrust of the review, so this error will only require some minor rewriting of the relevant passages.

Response: Thank you for pointing out the mistake. Some rewriting was done. In section 2.2 the text was changed into: CERV1 (GenBank acc. no. AY692036) and CERV2 (GenBank acc. no. AY692037) to search for CERV-like gammaretroviral sequences. In section 3.2 the text was changed into For instance, proviral integration times of chimpanzee CERV-K proviruses postdate the Homo-Pan lineage divergence (3.2.4), and ‘However, as more recent, lineage specific ERV-K insertions have been reported, the presence of specific ERV-K proviruses in OWM was investigated by querying their genome assemblies with full-length HERV-K and SERV-K1 sequences’, replaces the paragraph that included CERV1/2.

In the section on the SIVs, it is surprising that the author does not refer to the prosimian endogenous SIV of lemurs in the discussion of the age of the SIVs. It would seem that the phylogenetic relationship here would also contribute to understanding the time and origin of modern SIVs. It is also more evidence that retroviruses could migrate between bodies of land separated by water (ie a land bridge may not always be required).

Response: I did not discuss the topic as I felt it would complicate the story, but I agree that it adds to the understanding of SIV evolution. Therefore, added to page 5, paragraph 3.1.2 on SIV: Two independent integrations of an endogenous lentivirus, termed pSIV, with a phylogenetic position in between the feline lentivirus feline immunodeficiency virus (FIV) cluster and modern simian SIV strains, have been detected in the genome of Malagasy lemur species, suggesting that SIV precursors infecting primates existed already ~4.3 mya [83-86]. Since Madagascar and Africa have been separated for at least 160 mya, transitional pSIV was somehow transmitted across a substantial body of water, indicating that lentivirus dispersal does not always need a land bridge [83].

It could be clearer that SERV-K1/RhERV-K are phylogenetically part of the larger ERV-K or HERV-K (HML-2) clade, and that it has been well established that the oldest ERV-K proviruses in the human genome have orthologues in OWM (see ref 110).

Response: Added to the first line of paragraph 3.2.4 (page 11) describing endogenous betaretroviruses: The primate non-D type betaretrovirus superfamily, to which all ERV-K viruses belong,

Section 3.2 “Sequences homologous to HERV-K have been detected in all Old World primate species, with an estimated integration time of 28 mya” - The reference listed here is to a koala endogenous retrovirus paper that does not have any data regarding HERV-K.

Response: corrected, replaced by Reus et al, 2001 (ref. [121]).

Section 3.2.2 “So, as PcEV is present in Colobus, but not in cercopithecine monkeys…” The wording may be confusing to readers not familiar with primate taxonomy and nomenclature - the similarities of Cercopithecinidae, Cercopithecinae, Cercopithecini and in particular the informal term cercopithecine. Clarification of the informal term somewhere in the text would help the reader understand the argument being presented.

Response: Indeed, the names here are confusing. I have tried to make it better understandable by bolding the terms Cercopithecini and Papionini in Tables 1 & 2. Also, added to page 11, section 3.2.3 a further explanation has been added at the first occurrence of the term cercopithecine: in the cercopithecine species, that is, monkeys of the subfamily Cercopithecinae, thus both Cercopithecini and Papionini),

Page 6, last paragraph - Macaca should be italicized.

Response: Somehow the publisher submitted a different file for review than the one submitted, without line numbers and (most of the) italics. I have reinstated all italics in the revised file, hopefully it will stay that way. 

Page 7, the use of the term PTLV precedes the definition of the term.

Response: corrected on page 7 (PTLV (primate T-lymphotropic virus, the collective name for STLV and HTLV, the Homo-specific variants that arose after cross-species transmission).

The reference to PTLV-1 is somewhat confusing - is this a mistake, is it equivalent to STLV-1, or is it a collective term for HTLV-1 + STLV-1? Some clarification is needed here.

Response: Added to paragraph 3.1.4: Pathogenicity, both in humans and non-human primates, has convincingly been documented for PTLV-1 only (the cluster to which both STLV-1 and HTLV-1 belong,…)

Pages 7 and 8 - the viral genes “tax” and “env” should be italicized.

Response: Somehow the publisher submitted a different file for review than the one submitted, without line numbers and (most of the) italics. I have reinstated all italics in the revised file, hopefully it will stay that way. 

Page 8, near the top: “deltavirus” should be “deltaretrovirus”.

Response: corrected

Reviewer 2 Report

The Manuscript entitled „Contemporary distribution, estimated age and prehistoric migrations of Old World Monkey retroviruses by Antoinette C. van der Kuyl reviews and summarizes literature on the estimated age of exogenous retroviruses in the first section and continues to summarize literature on the presence proviral sequences in the germ line of OWM and their associated estimated integration events. Based on these insights the author discusses the age of 4 exogenous and puts that into the context in estimating the age of the active counterpart of now 4 known endogenous retroviruses. It is confusing that the article is submitted as a review but yet includes a result section and concludes with an own statement. That should be structured better in a revised version of the review.

Minor comments:

In a similar line it is stated that “this review from the first time includes evidence from extinct members of the family”. This statement goes beyond the function of a review and it is not clear what exactly the author is referring to.

The review should be more comprehensive on literature which compares retroviral integration sites among species with important insights on the pylogeneis among OWM and NWM.

The model does not take in to account that numerous restriction factors such as the TRIM and APOBEC enzymes block transmission among primates. should be discussed.

The prospect that some of the endogenous retroviruses also share similarities to retrotransposons especially in the context of foamy viruses is not discussed.

In Figure 1 I would suggest a better discrimination between exogenous and endogenous retroviruses (eg different colour in the border of the circles)

Author Response

Response to Reviewer 2 Comments

First, I am sorry that the template of the manuscript I have to submit for revision again does not include line numbers.

It is confusing that the article is submitted as a review but yet includes a result section and concludes with an own statement. That should be structured better in a revised version of the review.

Response: I had to choose the type of manuscript when uploading, but you are right, it is less of a review than a research article (same comment made by reviewer#3). I have added to the end of the Introduction that it is a ‘review and hypothesis paper’: This review and hypothesis paper will discuss, partly on the basis of new research, the epidemiology (who, where, when?) and putative age of primate retroviruses from the Oligocene till present, including both the extant and the extinct members of the family, to deduce probable (pre)historic OWM retrovirus dispersal routes.

Minor comments:

In a similar line it is stated that “this review from the first time includes evidence from extinct members of the family”. This statement goes beyond the function of a review and it is not clear what exactly the author is referring to.

Response: Replaced by including both the extant and the extinct members of the family.

The review should be more comprehensive on literature which compares retroviral integration sites among species with important insights on the pylogeneis among OWM and NWM.

Response: Investigating retroviral integration sites to help solve the phylogenetic relationships between species is a very interesting topic, but I fear it is outside the scope of the paper.

The model does not take in to account that numerous restriction factors such as the TRIM and APOBEC enzymes block transmission among primates. should be discussed.

Response: I agree. At first, I wanted to keep the analysis as simple as possible, but that is never the complete story. So, I have added to page 14, first paragraph: Of course, transmission of viruses, can be hampered by cellular and genetic variation between species. For SIV, immune system activity, including the activity of restriction factors such as TRIM5alpha and APOBEC3G, as well as amino acid variation in for instance the coreceptor protein, have all been implicated in the prevention of cross-species transmission (for a review, see [143]). Such inhibitory mechanisms could lead to false conclusions as to why certain species are free of infection, but would hardly play a role in conclusions drawn from infected species.

The prospect that some of the endogenous retroviruses also share similarities to retrotransposons especially in the context of foamy viruses is not discussed.

Response: Added to the first paragraph of the Introduction: Most likely, retroviruses originate from the long terminal repeat (LTR)-containing retrotransposons found in eukaryotic genomes, by the addition of an envelope (env) gene [6,7]. Indeed, one such LTR-retroelement, HERV-L, has similarity to foamy viruses in its polymerase (pol) gene [8].

In Figure 1 I would suggest a better discrimination between exogenous and endogenous retroviruses (eg different colour in the border of the circles)

Response: Figure 1 has been revised so that endogenous and exogenous viruses, depicted by circles, now have differently coloured borders.

Reviewer 3 Report

A pdf with detailed comments from the reviewer has been provided. 

Review Report for epidemiologia-1058034
General summary and comments –
In this very interesting review the author traces the association of retroviruses and various branches of the primate group using bioinformatics tools, anthropological data, and data from natural evolution studies. The author considers retrovirus infections which in both OWM and NWM and the timelines whereby these retrovirus genomes became integrated into primate genes. Along with a molecular clock like analogy, the author also elaborates on how the viruses have travelled with their hosts and spread to multiple continents during the time period roughly from Late Oligocene up to the Holocene. While the review is extensive and undeniably interesting, it needs further edits before it can be accepted for publication.
Minor comments –
1) There are some typographical errors. I highly recommend a quick read through which should be enough to clean them up. Pay close attention to sentence construction, prepositions and conjunctions. 2) I am not sure if this was the choice of the editor or the author but the review/manuscript seems to be in its final formatted form. No double line spacing or line numbers are on any page making very difficult to give comments. 3) Is this a review or a manuscript? Maybe this is a new hybrid type of journal article I have not come across before where a “review” has experiments and results also.
Major Comments –
1) Scientific names of the OWM and NWM are not italicized. This has become an erroneous and common practice but I literally lose my mind when someone does this. Genus and species are still written italicized unless the international convention has changed. 2) How many Figure 1s are there in this manuscript? How are figures divided between full figures and Appendices? Figures consisting of phylogenetic trees seem low resolution. Is this a result of converting the image to pdf? 3) Another comment is about the presence of both apes and monkeys in Eurasia, actually mainly in Europe in the mid Miocene from 15 MYA to 6 MYA. Eg – Members of the Oreopithecus or Morotopithecus genera. Is there any data on their genomes, viruses that infected them and any endogenous or exogenous retroviruses that the author could talk about? While all these apes became extinct towards the end of Oligocene, they still were reported to have existed in Europe for about 8-10 million years. Is there any data available on these primates and the related retroviruses?

Line by line comments –
1) Page 2, Materials and Methods, Section 2.1 – Was the literature search done manually or by an algorithm?
2) Page 3, Materials and Methods, Section 2.2 - Since we are talking about extinct species, how complete were these genomes? What fraction of the genome was analyzed for analyses? Provide a table for reference listing the extant and extinct primates. Why are none of the genus and species names italicized? 3) Page 3, Materials and Methods, Section 2.3 - Provide more details of this section. Looks like you definitely did more than just align sequences. Your figures indicate phylogeny trees with outgroups and all constructed. Even if these are default results, please describe ALL analyses you performed. 4) Page 4, Results, Section 3.1.1 - Are there noncoding parts of retroviral genomes? 5) Page 6, Results, Section 3.1.2 – Reference 84 in your list – where the author talks about primate settlements from Europe and Arabia bridging the gap between Africa and Asia. I highly recommend watching the following Youtube video. In the description, they list several dozen primary literature articles - https://www.youtube.com/watch?v=qzRzUmrHL7c. Please see if you can use any of those literature articles and comment or elaborate on my Major Comment #3. 6) Page 7, Section 3.1.3, Figure 1 - While this is still a cartoon and fully acceptable for representative purposes, does this map represent the continents and land masses as they are for the ~56 to 3 mya time period? Shouldn’t the Americas be closer to Africa, the Subcontinent plate shaped slightly differently, even the continent of Antarctica was placed differently. Was the Isthmus of Panama formed during this time? Or the Bering Land Bridge - specifically since your most recent virus integration/speciation estimated do cover the last ice age. 7) Page 7, Section 3.1.3, “Often, studies are based on” - Transmission of STLV to Humans happened between 27 to 8 yrs ago? Are we missing a unit or a significant number here? This reference paper was published in 2001. 8 yrs ago was 2012-2013. Which means this 2001 paper predicted the transmission of STLV to humans ~10 yrs before it actually happened? I guess this is a typo but makes it look scientifically incoherent. Also underscores my point of having line numbers in manuscripts sent for review. 8) Page 8, Section 3.2 - Is there any consensus as to why would a retrovirus integrate into the germline of a host vs why it wouldn’t? More of a curiosity question 9) Page 11, Results, Section 3.2.4 - Was there an equitable population distribution of all host primate species considered all across the Late Oligocene to early Miocene?

Author Response

Response to Reviewer 3 Comments

First, I am sorry that the template of the manuscript I have to submit for revision again does not include line numbers.

Minor comments –
1) There are some typographical errors. I highly recommend a quick read through which should be enough to clean them up. Pay close attention to sentence construction, prepositions and conjunctions.

Response: The manuscript was edited to improve English grammar.

2) I am not sure if this was the choice of the editor or the author but the review/manuscript seems to be in its final formatted form. No double line spacing or line numbers are on any page making very difficult to give comments. 3) Is this a review or a manuscript? Maybe this is a new hybrid type of journal article I have not come across before where a “review” has experiments and results also.

Response: Yes, it is the policy of the journal to submit manuscripts using a formatted template. And I had to choose the type of manuscript when uploading, but you are right, it is less of a review than a research article. I have added to the end of the Introduction that it is a ‘review and hypothesis paper’: This review and hypothesis paper will discuss, partly on the basis of new research, the epidemiology (who, where, when?) and putative age of primate retroviruses from the Oligocene till present, including both the extant and the extinct members of the family, to deduce probable (pre)historic OWM retrovirus dispersal routes.

Major Comments –
1) Scientific names of the OWM and NWM are not italicized. This has become an erroneous and common practice but I literally lose my mind when someone does this. Genus and species are still written italicized unless the international convention has changed.

Response: Somehow the publisher submitted a different file for review than the one submitted, without line numbers and (most of the) italics. I have reinstated all italics in the revised file, hopefully it will stay that way. 

2) How many Figure 1s are there in this manuscript? How are figures divided between full figures and Appendices? Figures consisting of phylogenetic trees seem low resolution. Is this a result of converting the image to pdf?

Response: When uploading extra material for this journal, it could either be as an appendix (at the end of the paper) or as a supplementary file. As I myself do not like to have to go back to a website to get supplementary files, I decided to add the phylogenetic trees, which are not essential for the main message, as appendices. So, there are two figures in the main text, and two figures in the appendix (one with trees based on full-length sequences, the other on LTRs only).

3) Another comment is about the presence of both apes and monkeys in Eurasia, actually mainly in Europe in the mid Miocene from 15 MYA to 6 MYA. Eg – Members of the Oreopithecus or Morotopithecus genera. Is there any data on their genomes, viruses that infected them and any endogenous or exogenous retroviruses that the author could talk about? While all these apes became extinct towards the end of Oligocene, they still were reported to have existed in Europe for about 8-10 million years. Is there any data available on these primates and the related retroviruses?

Response: It would be great should anything be available on the genome of these very interesting primate species. Unfortunately, ancient DNA sequencing cannot yet be applied to these early times. Maybe a next generation of researchers will be able to unravel the secrets of the European apes and monkeys from the Miocene!

Line by line comments –
1) Page 2, Materials and Methods, Section 2.1 – Was the literature search done manually or by an algorithm?

Response: Manually Added to M&M section 2.1 on page 2.

2) Page 3, Materials and Methods, Section 2.2 - Since we are talking about extinct species, how complete were these genomes? What fraction of the genome was analyzed for analyses? Provide a table for reference listing the extant and extinct primates. Why are none of the genus and species names italicized?

Response: I do not understand the remark, as no genomes of extinct species were analysed as none are available (except for Homo, which was not studied here). And somehow, the publisher submitted a different file for review than the one submitted, without line numbers or (most of the) italics. I have restored all italics in the revised file, hopefully it will stay that way.

3) Page 3, Materials and Methods, Section 2.3 - Provide more details of this section. Looks like you definitely did more than just align sequences. Your figures indicate phylogeny trees with outgroups and all constructed. Even if these are default results, please describe ALL analyses you performed.

Response: As the phylogenetic analyses were only shown in the Appendices, with a description of the methods used in the legend of the figures, I wasn’t sure whether or not to include that information in the M&M section. But now I have added to section 2.3 on page 3: Sequence distances were calculated with the Kimura-2-parameter method and evolutionary relationships were inferred using the Neighbor-joining method with bootstrapping as implemented in MEGA6 [40,41]. Gaps/missing data treatment was set to ‘partial deletion’ with a cut-off value of 90%. No outgroup was defined.

4) Page 4, Results, Section 3.1.1 - Are there noncoding parts of retroviral genomes?

Response: Yes, the LTRs of retroviruses are non-coding, and should be available. However, when analysing ancient relationships, coding parts might be more informative, as non-coding sequences tend to mutate faster

5) Page 6, Results, Section 3.1.2 – Reference 84 in your list – where the author talks about primate settlements from Europe and Arabia bridging the gap between Africa and Asia. I highly recommend watching the following Youtube video. In the description, they list several dozen primary literature articles - https://www.youtube.com/watch?v=qzRzUmrHL7c. Please see if you can use any of those literature articles and comment or elaborate on my Major Comment #3.

Response: It is a great video, and makes one interested in writing on ape retroviruses! Regrettably, nothing is known on any viruses, endogenous or infectious, bothering these species. Should we have endogenous sequences, those could also be used to infer phylogenetic relationships between extinct and extant primates. I do hope ancient DNA studies will be able to use much older material in the future.

6) Page 7, Section 3.1.3, Figure 1 - While this is still a cartoon and fully acceptable for representative purposes, does this map represent the continents and land masses as they are for the ~56 to 3 mya time period? Shouldn’t the Americas be closer to Africa, the Subcontinent plate shaped slightly differently, even the continent of Antarctica was placed differently. Was the Isthmus of Panama formed during this time? Or the Bering Land Bridge - specifically since your most recent virus integration/speciation estimated do cover the last ice age.

Response: No, the figure represent the continents as they are now. I considered doing a time series, but that would not be more informative, especially as infectious periods and travels of the viruses have been estimated and are not absolute. Added to the legend of Fig. 1: The continental positions shown represent the current situation, the time periods shown are estimates. During the Miocene, a junction existed between Africa and Asia due to the closure of the Tethys Seaway.

7) Page 7, Section 3.1.3, “Often, studies are based on” - Transmission of STLV to Humans happened between 27 to 8 yrs ago? Are we missing a unit or a significant number here? This reference paper was published in 2001. 8 yrs ago was 2012-2013. Which means this 2001 paper predicted the transmission of STLV to humans ~10 yrs before it actually happened? I guess this is a typo but makes it look scientifically incoherent. Also underscores my point of having line numbers in manuscripts sent for review.

Response: Sorry, it was indeed a typo as 27,300 ± 8,200 years ago was meant in section 3.1.4. And the line numbers were present in the submitted manuscript, but have apparently been removed before sending it out for review.

8) Page 8, Section 3.2 - Is there any consensus as to why would a retrovirus integrate into the germline of a host vs why it wouldn’t? More of a curiosity question

Response: Not many people have apparently given it a thought, but likely a retrovirus does no target such cells, but just happens to be able to infect them. Some have the ability to reach the genital tract, or infect the egg or sperm cell, especially when their receptor is expressed. For some ideas, see: van der Kuyl & Berkhout, https://doi.org/10.1016/j.virusres.2020.198101

9) Page 11, Results, Section 3.2.4 - Was there an equitable population distribution of all host primate species considered all across the Late Oligocene to early Miocene?

Response: I guess we don’t know, we can only do the analysis in hindsight. Such a lack of information is of course a weakness of the suggested virus migration routes.